# Characterization of the Synergistic Inhibition of *I*_K(erg)_ and *I*_K(DR)_ by Ribociclib, a Cyclin-Dependent Kinase 4/6 Inhibitor

**DOI:** 10.3390/ijms21218078

**Published:** 2020-10-29

**Authors:** Pin-Yen Liu, Wei-Ting Chang, Sheng-Nan Wu

**Affiliations:** 1Division of Cardiovascular Medicine, Department of Internal Medicine, College of Medicine, National Cheng Kung University Hospital, Tainan 704, Taiwan; larry@mail.ncku.edu.tw; 2Institute of Clinical Medicine, College of Medicine, National Cheng Kung University, Tainan 704, Taiwan; cmcvecho2@gmail.com; 3Division of Cardiovascular Medicine, Chi-Mei Medical Center, Tainan 710, Taiwan; 4Department of Biotechnology, Southern Taiwan University of Science and Technology, Tainan 71005, Taiwan; 5Department of Physiology, National Cheng Kung University Medical College, Tainan 704, Taiwan; 6Institute of Basic Medical Sciences, National Cheng Kung University Medical College, Tainan 704, Taiwan

**Keywords:** delayed-rectifier potassium current, erg-mediated potassium current, heart cell, M-type potassium current, pituitary cell, ribociclib, voltage hysteresis

## Abstract

Ribociclib (RIB, LE011, Kisqali^®^), an orally administered inhibitor of cyclin-dependent kinase-4/6 (CDK-4/6) complex, is clinically effective for the treatment of several malignancies, including advanced breast cancer. However, information regarding the effects of RIB on membrane ion currents is limited. In this study, the addition of RIB to pituitary tumor (GH_3_) cells decreased the peak amplitude of *erg*-mediated K^+^ current (*I*_K(erg)_), which was accompanied by a slowed deactivation rate of the current. The IC_50_ value for RIB-perturbed inhibition of deactivating *I*_K(erg)_ in these cells was 2.7 μM. In continued presence of μM RIB, neither the subsequent addition of 17β-estradiol (30 μM), phorbol 12-myristate 13-acetate (10 μM), or transforming growth factor-β (1 μM) counteracted the inhibition of deactivating *I*_K(erg)_. Its presence affected the decrease in the degree of voltage-dependent hysteresis for *I*_K(erg)_ elicitation by long-duration triangular ramp voltage commands. The presence of RIB differentially inhibited the peak or sustained component of delayed rectifier K^+^ current (*I*_K(DR)_) with an effective IC_50_ of 28.7 or 11.4 μM, respectively, while it concentration-dependently decreased the amplitude of M-type K^+^ current with IC_50_ of 13.3 μM. Upon 10-s long membrane depolarization, RIB elicited a decrease in the *I*_K(DR)_ amplitude, which was concomitant with an accelerated inactivation time course. However, the inability of RIB (10 μM) to modify the magnitude of the hyperpolarization-activated cation current was disclosed. The mean current–voltage relationship of *I*_K(erg)_ present in HL-1 atrial cardiomyocytes was inhibited in the presence of RIB (10 μM). Collectively, the hyperpolarization-activated cation current was observed. RIB-mediated perturbations in ionic currents presented herein are upstream of its suppressive action on cytosolic CDK-4/6 activities and partly participates in its modulatory effects on the functional activities of pituitary tumor cells (e.g., GH_3_ cells) or cardiac myocytes (e.g., HL-1 cells).

## 1. Introduction

Ribociclib (RIB, LE011, Kisqali^®^) is an oral small-molecule inhibitor of cyclin-dependent kinase-4/6 (CDK-4/6) complex. It has been used to treat certain types of breast (e.g., hormone receptor-positive, HER2-negative breast cancer) or prostate cancers [1,2,3,4,5]. Additionally, this compound has been studied as a treatment for other drug-resistant cancers or neuroendocrine tumors that include pituitary tumors [3,6,7,8,9,10].

CDK inhibitors (e.g., RIB) are thought to target and deregulate different types of tumor cells, including pituitary tumor cells. [7,9,11,12,13]. An earlier report revealed the effectiveness of palbociclib, another CDK-4/6 inhibitor, in inducing the regression of pituitary microadenoma [7]. CDK-4/6 inhibitors (e.g., RIB) protect motor neurons from degenerative changes in an organoid spinal muscular atrophy model [3]. Of note, RIB has been recently revealed to produce a significant prolongation of the electrocardiographic QTc interval, as it was used to treat patients with advanced cancer [14,15]. However, it was well tolerated without any unanticipated adverse effects. However, it is not clear whether RIB treatment causes any direct perturbations in the magnitude of different types of plasmalemmal ionic currents.

The *erg*-mediated K^+^ current (*I*_K(erg)_) is known to be encoded by three different subfamilies of *KCNH*, and it can give rise to the pore-forming α–subunit of *erg*-mediated K^+^ (i.e., K_erg_ or K_V_11) channels [16]. This current is regarded as constituting the cloned counterpart of the rapidly activating delayed-rectifying K^+^ currents in heart cells [17]. The *I*_K(erg)_ is inherent in neurons or different types of electrically excitable cells, such as endocrine or neuroendocrine cells. It can significantly affect the maintenance of the resting membrane potential and the modifications in sub-threshold excitability [18,19,20], thereby leading to significant changes in the discharge rate of spontaneous action potentials [21,22]. Consequently, the resultant perturbations in these KV channels’ activity have the propensity to modify the stimulus-secretion coupling of these cells [23]. Previous work has also revealed that the magnitude of *I*_K(erg)_ can be a regulator of tumor cell apoptosis and proliferation [24,25,26]. However, whether the presence of RIB produces any modifications of the magnitude or gating in this type of ion channel is unknown.

Thus, we examined whether RIB produces any perturbations of the various types of ionic currents through the membranes of excitable cells, including *I*_K(erg)_, delayed-rectifier K^+^ current (*I*_K(DR)_), M-type K^+^ current (*I*_K(M)_), and hyperpolarization-activated cation current (*I*_h_) found in pituitary tumor (GH_3_) cells. We also evaluated the possible effect of RIB on *I*_K(erg)_ in HL-1 cardiomyocytes. This study revealed that RIB could decrease *I*_K(erg)_ and *I*_K(DR)_, the ionic mechanisms that do not appear to have a causal link to its suppression of CDK-4/6 activity. The importance of RIB-induced inhibition of CDK-4/6 activity seems to be overestimated, notwithstanding the presence of the CDK complex in the cytosol of pituitary tumor cells or heart cells [9,27,28,29].

## 2. Materials and Methods

### 2.1. Drugs, Chemicals, and Solutions Prepared for This Study

Ribociclib (RIB, LEE011, Kisqali^®^, 7-cyclopentyl-*N*,*N*-dimethyl-2-[(5-piperazin-1-ylpyridin-2-yl)amino]pyrrolo[2,3-d]pyrimidine-6-carboxamide, C_23_H_30_N_8_O, PubMed CID: 44631912) was acquired from MedChemExpress (Everything Biotech Ltd., Taipei, Taiwan). XE991 was purchased from Tocris (Union Biomed Inc., Taipei, Taiwan), E-4031 was purchased from Enzo (Blossom Biotechnologies, Taipei, Taiwan), and 17β-estradiol, lamotrigine, norepinephrine, phorbol 12-myristate 13-acetate (PMA), tetraethylammonium chloride (TEA), tetrodotoxin, and transforming growth factor-β were from Sigma-Aldrich (Merck Ltd., Taipei, Taiwan). Tissue culture media, horse and fetal calf serum, L-glutamine, and trypsin/EDTA were acquired from HyClone™ (Thermo Fisher; Level Biotech, Tainan, Taiwan), and other chemicals—including CdCl_2_, aspartic acid, and HEPES—were of the best available quality, mostly at analytical grade. Deionized water used in all experiments was obtained from a Milli-Q water purification system (Merck Millipore, Taipei, Taiwan).

The composition of HEPES-buffered normal Tyrode’s solution was 136.5 mM NaCl, 5.4 mM KCl, 1.8 mM CaCl_2_, 0.53 mM MgCl_2_, 5.5 mM glucose, and 5.5 mM HEPES-NaOH buffer, pH 7.4. To record *I*_K(DR)_ or *I*_h_, the patch electrode was filled with the following solution: 140 mM KCl, 1 mM MgCl_2_, 3 mM Na_2_ATP, 0.1 mM Na_2_GTP, 0.1 mM EGTA, and 5 mM HEPES-KOH buffer, pH 7.2. To measure *I*_K(erg)_ or *I*_K(M)_, the bath solution was replaced with a high-K^+^, Ca^2+^-free solution containing 130 mM KCl, 10 mM NaCl, 3 mM MgCl_2_, and 5 mM HEPES-KOH buffer, pH 7, while the internal pipette solution contained 145 mM KCl, 2 mM MgCl_2_, and 5 mM HEPES-KOH buffer, pH 7.2. On the day of the experiments, each solution was filtered with a 0.22-µm pore size sterile syringe filter (Bio-Check, New Taipei City, Taiwan).

### 2.2. Cell Preparations

The GH_3_ clonal cell line derived from rat anterior pituitary cells, acquired from the Bioresources Collection and Research Center ([BCRC-60015]; Hsinchu, Taiwan), was cultured in Ham’s F-12 medium supplemented with 15% (*v*/*v*) horse serum, 2.5% (*v*/*v*) fetal calf serum and 2 mM L-glutamine [30,31]. The HL-1 atrial cell line was derived from the AT-1 mouse atrial cardiomyocyte tumor lineage, acquired initially from Louisiana State University in New Orleans, LA. Cells were grown in Claycomb medium (Sigma-Aldrich, St. Louis, MO, USA) supplemented with 10% fetal bovine serum, 100 μM norepinephrine, and 2 mM L-glutamine [32]. These cells were maintained at 37 °C in a humidified atmosphere containing 95% air and 5% CO_2_. The viability of GH_3_ or HL-1 cells used in this study was determined using the trypan blue dye exclusion test. For subculturing, cells were trypsin-dissociated and passaged every 2–3 days. The electrical recordings were undertaken five or six days after cells had been cultured (60–80% confluence).

### 2.3. Electrophysiological Measurements

Briefly, before the measurements were prepared, GH_3_ or HL-1 cells were harvested with 1% trypsin/EDTA solution, and a cell suspension aliquot was rapidly transferred to a home-made recording chamber that was firmly mounted on the stage of an inverted DM-IL microscope (Leica, Major Instruments, New Taipei City, Taiwan). The cells studied were suspended at room temperature (20–25 °C) in HEPES-buffered normal Tyrode’s solution, the components of which are stated above. The cells were then allowed to settle to the bottom of the chamber. To fabricate the patch electrode from Kimax-51 capillaries (#34500; Kimble: Dogger, New Taipei City, Taiwan), we used either a PP-830 two-step puller (Narishige: Taiwan Instrument Co., Taipei, Taiwan) or a P-97 Flaming/Brown horizontal programmable puller (Sutter, Novato, CA, USA). We then fire-polished the tips with an MF-83 microforge (Narishige). During the measurements, the electrodes, which bore resistances between 3 and 5 MΩ when filled with different internal solutions, were maneuvered by using an MX-4 manipulator (Narishige) and delicately operated by an MHW-3 hydraulic micromanipulator (Narishige). We performed standard patch-clamp recordings in a whole-cell configuration using either an Axoclamp-2B (Molecular Devices, Sunnyvale, CA, USA) or an RK-400 (Bio-Logic, Claix, France) amplifier [33,34]. Consistent with previous observations (Milton and Caldwell, 1990), the formation of a bleb of membrane lipids in the electrode tip based on microscopic observation of seal formation was also noticed. As the electrodes with membrane seals were measured to be less than 1 GΩ, the recordings were discarded. The junctional potentials, which develop at the electrode tip when the composition of the internal solution differed from that in the bath, were nulled before the start of each giga-seal formation, and such junction potentials corrected whole-cell data. The tested compounds were applied through perfusion or were added to the bath to achieve the final concentration indicated.

### 2.4. Data Recordings

The signals (i.e., the voltage and current traces in response to command pulses) were simultaneously monitored on either an HM-507 (Hameg, East Meadow, NY, USA) or a Hantek-6022BC oscilloscope (Qingdao, Shangdong, China). The signals were digitally sampled and stored at 10 kHz in an ASUSPRO-BU401LG computer (ASUS, Tainan, Taiwan). Data acquisition, to which the Digidata 1440A (Molecular Devices) was connected, was operated using pCLAMP 10.7 software (Molecular Devices). During the recordings, the signals were also monitored on a liquid crystal display monitor (MB169B+; ASUS, Taipei, Taiwan) through a universal series bus (USB) type-C connection. The laptop computer used was placed on the top of an adjustable Cookskin stand (Ningbo, Zhejiang, China) for efficient manipulation. The acquired current signals were low-pass filtered at 2 kHz through an FL four-pole Bessel filter (Dagan, Minneapolis, MN, USA) to minimize electrical noise. Through digital-to-analog conversion, the pCLAMP-generated voltage profiles comprising various types of rectangular or isosceles-triangular waveforms were designed and delivered to assess the current-voltage (*I–V*) relationships of different ionic currents or voltage hysteresis of the specified currents. To ensure digitalization, some of the experiments were acquired using a PowerLab 2/26 acquisition system (AD Instruments; KYS Technology, Tainan, Taiwan).

### 2.5. Data Analyses

To determine the percentage of inhibition by RIB on *I*_K(erg)_ or *I*_K(M)_ amplitude, cells were bathed in high-K^+^, Ca^2+^-free solution or Ca^2+^-free Tyrode’s solution, respectively; for the *I*_K(DR)_ amplitude, cells were suspended in Ca^2+^-free, Tyrode’s solution. To measure *I*_K(erg)_ or *I*_K(M)_, each cell was either hyperpolarized from −10 to −90 mV or depolarized from −50 to −10 mV, respectively. To record *I*_K(DR)_, the examined cell was abruptly depolarized from −50 to +50 mV. The *I*_K(erg)_, *I*_K(DR)_, or *I*_K(M)_ amplitudes during the application of RIB were compared with those measured after subsequent addition of E-4031 (10 μM), tetraethylammonium chloride (TEA; 10 mM), or XE991 (10 μM), respectively. E-4031 or XE991 is known to be a selective blocker of *I*_K(erg)_ or *I*_K(M)_, respectively [21,35], and TEA can effectively suppress the amplitude of *I*_K(DR)_. The concentration-dependent relationships of RIB on the inhibition of *I*_K(erg)_, *I*_K(DR)_ (peak and sustained components), or *I*_K(M)_ amplitude were fit by a three-parameter logistic model (i.e., modified Hill function). That is,
$$\mathrm{Percentage}\;\mathrm{inhibition}\;\left(\%\right)=\frac{E_{\max}\times {\left[\mathrm{RIB}\right]}^{n_{H}}}{{\mathrm{IC}}_{50}^{n_{H}}+{\left[\mathrm{RIB}\right]}^{n_{H}}}$$
where IC_50_ or n_H_ is defined as the 50% inhibitory concentration (i.e., the concentration corresponding to the response midway between the estimates for the minimum and the maximum) or the Hill coefficient, respectively; RIB is the ribociclib concentration; and E_max_ indicates the maximal inhibition of *I*_K(erg)_ (i.e., E-4031-sensitive current), *I*_K(DR)_ (i.e., TEA-sensitive current), or *I*_K(M)_ (i.e., XE991-sensitive current) amplitude produced by the presence of RIB.

### 2.6. Statistical Analyses

Linear or nonlinear regression analyses for various data sets (e.g., the logistic model and the time-course for the trajectory of the current activation, inactivation, or deactivation) were obtained from the goodness of fit using different analytical methods, such as the Microsoft Excel-embedded “Solver” function (Redmond, WA, USA) and OriginPro 2016 (OriginLab; Schmidt Scientific, Kaohsiung, Taiwan). The experimental results are expressed as the mean ± standard error of the mean (SEM). Sample sizes (n) are the cell number from which the data in a separate set of experiments were collected; error bars are plotted as one SEM. A paired or unpaired *t*-test followed by a one-way analysis of variance (ANOVA), together with the least significant difference method for multiple-group comparisons among means, were used for statistical evaluations. The analyses were performed using SPSS version 20.0 (Asia Analytics, Taipei, Taiwan). Differences between values were considered significant when *p* < 0.05.

## 3. Results

### 3.1. Effect of RIB on Erg-Mediated K^+^ Current (I_K(erg)_) Identified in Pituitary GH_3_ Cells

In the initial stage of the experiments, whole-cell voltage-clamp recordings were used to measure the perturbations of RIB on plasmalemmal ionic currents. To record *I*_K(erg)_ and avoid possible contamination of Ca^2+^-activated K^+^ currents, we suspended cells in a high-K^+^, Ca^2+^-free solution and then filled the recording electrode using a K^+^-enriched internal solution. The cell was voltage-clamped at −10 mV, and 1-s hyperpolarizing voltage commands to −90 mV was then applied to evoke *I*_K(erg)_ [36,37,38]. As cells were exposed to different concentrations of RIB, the *I*_K(erg)_ responding to 1-s membrane hyperpolarization progressively decreased, and the deactivation time-course of the current concomitantly became accelerated (Figure 1A). For example, when we hyperpolarized to −90 mV from a holding potential of −10 mV, RIB at a concentration of 3 μM produced a considerable reduction in the peak amplitude (i.e., measured at the beginning of the hyperpolarizing pulse) of deactivating *I*_K(erg)_ from 589 ± 23 to 368 ± 18 pA (n = 8, *p* < 0.05). The presence of RIB (3 μM) also produced an increase in the deactivating time constant (τ_deact_) from 349 ± 21 to 524 ± 31 ms (n = 7, *p* < 0.05). After the compound was washed out, the current amplitude returned to 246 ± 21 pA (n = 7, *p* < 0.05). Figure 1B shows the time-course of RIB-induced changes in *I*_K(erg)_ amplitude activated in response to step hyperpolarization from −10 to −90 mV is illustrated. Additionally, neither the further addition of 17β-estradiol (30 μM), phorbol 12-myristate 13-acetate (PMA, 10 μM), or transforming growth factor-β (1 μM) reversed RIB-mediated inhibition of *I*_K(erg)_. Furthermore, 17 β-Estradiol, PMA, or transforming growth factor-β was previously reported to affects CDK-4/6 complex activities [39,40,41].

We further assessed the relationship between RIB concentration and the percentage of inhibition of *I*_K(erg)_. The amplitudes of *I*_K(erg)_ in the presence of RIB were compared with those after the subsequent addition of E-4031 (10 µ µM). E-4031 is reportedly a selective inhibitor of *I*_K(erg)_ [21]. According to a four-parameter logistic model (i.e., modified Hill equation) described in the Materials and Methods and using a nonlinear least-squares fit to the experimental results, the IC_50_ value needed for the inhibitory effect of RIB on *I*_K(erg)_ amplitude in GH_3_ cells was calculated to be 2.7 µM, and at a concentration of 100 µM, this compound almost entirely inhibited the current amplitude (Figure 1C). The experimental results indicate that *I*_K(erg)_ remained functionally active in GH_3_ cells [21,23] and that RIB is capable of exerting a modulatory action on the magnitude or time-course of deactivating *I*_K(erg)_ produced by membrane hyperpolarization.

### 3.2. Effect of RIB on the Mean Current–Voltage (I–V) Relationship of I_K(erg)_

Next, we determined RIB’s impact on the *I–V* relationships of *I*_K(erg)_ in GH_3_ cells. Figure 2A illustrates representative current traces obtained after applying 1-s hyperpolarizing steps from a holding potential of −10 mV to membrane potentials ranging between −120 and −10 mV in 10−mV steps taken without or with the addition of RIB (10 μM). The magnitude of the RIB-mediated inhibitory effect on *I*_K(erg)_ measured at the beginning of the hyperpolarizing voltage commands noticeably became higher than that at the end of the voltage step (Figure 2B,C). The presence of RIB (10 μM) led to a resultant decrease in the slope of the linear fit of the peak *I*_K(erg)_ amplitude to the voltages between −120 and −90 mV from 10.7 ± 1.4 to 3.9 ± 0.8 nS (n = 7, *p* < 0.05). These data strongly indicate that RIB’s blocking effect was mainly exerted on *I*_K(erg)_, which is responsible for inward rectification [36].

### 3.3. Effect of RIB on the Voltage Hysteresis of I_K(erg)_ Elicited during an Isosceles-Triangular Ramp Pulse

The voltage-dependent hysteresis of various ionic currents has been demonstrated to exert significant impacts on electrically excitable cells [38,42,43]. Thus, we further explored whether RIB’s presence was capable of perturbing voltage hysteresis inherently existing in *I*_K(erg)_ recorded from GH_3_ cells. In this series of measurements, a long-lasting isosceles-triangular ramp pulse with a duration of 6 s (i.e., ±0.53 V/s) was digitally created and then delivered to the cell through digital-to-analog conversion, as the whole-cell configuration was firmly achieved. Of particular interest, as depicted in Figure 3, the trajectories of *I*_K(erg)_ responding to the upsloping (i.e., ramp-depolarization from −160 to 0 mV) and downsloping (ramp-hyperpolarization from 0 to −160 mV) ramp pulse as a function of time formed a loop and were virtually distinguishable from each other. Alternatively, the current amplitude in response to the upsloping (ascending) limb of the triangular voltage ramp was greater than that of the downsloping (descending) limb. This observation (i.e., the relationship of *I*_K(erg)_ amplitude versus membrane potential) indicates a voltage-dependent hysteresis for this current, as highlighted in Figure 3 [38]. Moreover, as the examined cell was continually exposed to RIB (10 μM), the *I*_K(erg)_ amplitude evoked in the upsloping or downsloping limb of the long-lasting triangular ramp was reduced to a similar extent. For example, in the control, the *I*_K(erg)_ amplitude measured at −100 mV in response to the upward or downward end of the triangular ramp pulse was 131 ± 15 or 71 ± 11 pA (n = 8), respectively, and the values were found to differ significantly between them (*p* < 0.05). However, in the presence of 10 μM RIB, the amplitude of the forward or backward *I*_K(erg)_ measured at the same level of membrane potential was diminished to 41 ± 9 or 11 ± 4 pA (n = 8, *p* < 0.05), respectively.

Next, we quantified the strength of voltage hysteresis based on the difference in the area (i.e., Δarea) under the curve in the upsloping (forward) and downsloping (backward) direction, as indicated by the arrow and shaded area in Figure 3A. We noticed from these observations that for *I*_K(erg)_ identified in GH_3_ cells, the degree of voltage hysteresis increased with slower ramp speed and that cell exposure to RIB led to a considerable reduction in the amount (i.e., Δarea) of such hysteresis. Figure 3B shows a summary of the experimental results demonstrating RIB’s suppressive effect on the area-under-the-curve between the forward and backward current traces. For example, the addition of 3 or 10 μM RIB decreased the area by about 50% or 80% elicited by such a long-lasting triangular ramp pulse.

### 3.4. Effect of RIB on Delayed-Rectifier K^+^ Current (I_K(DR)_) Recorded from GH_3_ Cells

In the next stage of the experiments, we examined whether RIB produces any inherent perturbations on *I*_K(DR)_ in these cells. To minimize the contamination of Ca^2+^-activated K^+^ currents, we kept cells bathed in Ca^2+^-free Tyrode’s solution, and the recording electrode was filled with K^+^-containing internal solution. The compositions of these solutions are detailed above. As the whole-cell configuration of the patch-clamp recordings was firmly established, we voltage-clamped the examined cell at −50 mV, and 1 s depolarizing voltages ranging between −50 and +50 mV were applied to evoke K^+^ currents. As shown in Figure 4A–C, within 1 min of exposing the cell to 10 μM RIB, the amplitudes of *I*_K(DR)_ measured at voltages ranging between 0 and +50 mV were considerably reduced. Moreover, the decaying (or relaxation) time-course of the current in response to sustained depolarization became faster in the presence of RIB. The mean *I–V* relationship of the sustained *I*_K(DR)_ amplitude with or without cell exposure to 10 μM RIB is illustrated in Figure 4B or Figure 4C, respectively. The concentration-dependent relationships of the effect of RIB on the amplitude of *I*_K(DR)_ (i.e., TEA-sensitive current) measured either at the beginning (i.e., peak current) or the end (i.e., sustained current) of the depolarizing command step from −50 to +50 mV were derived and are plotted in Figure 4D. The half-maximal inhibitory concentration (IC_50_) required for RIB-perturbed suppression of peak or sustained *I*_K(DR)_ in GH_3_ cells was reliably estimated to be 28.7 or 11.4 μM, respectively. However, a minimal change in the Hill coefficient of the relationship was noticed. Therefore, owing primarily to the elevation of the inactivation time course responding to maintained depolarization, the IC_50_ for the RIB-induced decrease of *I*_K(DR)_ (i.e., TEA-sensitive current) amplitude was observed to be approximately 2.5-fold less in the current magnitude measured at the beginning of the depolarizing step than that at the end of the pulse.

### 3.5. Effect of RIB on 10-s Long Depolarization-Elicited I_K(DR)_

When cells were depolarized for 10 s, a considerable reduction in *I*_K(DR)_ inactivation could be readily demonstrated. Hence, it was noticed that the time-course of inactivation in this type of K^+^ current tended to last tens of seconds. Under this experimental protocol, the state-dependent inactivation (i.e., decreased channel-opening availability) of the *I*_K(DR)_ channel could be thoroughly investigated. Thus, we further characterized the *I*_K(DR)_ in response to 10-s depolarization without or with exposure to RIB. As cells were exposed to RIB, the *I*_K(DR)_ amplitude measured at the end of a 10-s depolarization from −50 to +50 mV progressively decreased. Moreover, the inactivation time constant (τ_inact_) of the current was significantly elevated in the presence of RIB (Figure 5). For example, the presence of 3 μM RIB consistently decreased the *I*_K(DR)_ amplitude at the end of the depolarizing step from 110 ± 19 to 54 ± 11 pA (n = 8, *p* < 0.05); and concomitantly with this observation, the value of τ_inact_ was significantly increased to 1.92 ± 0.05 s (n = 8, *p* < 0.05) from a control of 1.67 ± 0.04 s. Therefore, it is possible that the slowly inactivating K^+^ conductance perturbed by the presence of RIB results in delayed firing or long-term potentiation of action potentials in electrically excitable cells [30,44,45].

### 3.6. Effect of RIB on M-Type K^+^ Current (I_K(M)_) in GH_3_ Cells

We continued to evaluate whether RIB could alter another type of K^+^ current, *I*_K(M)_ [31]. This set of whole-cell current recordings was conducted in cells suspended in high-K^+^, Ca^2+^-free solution, and the recording pipette was filled with K^+^-containing internal solution. However, distinguishable from its modulatory effects on *I*_K(erg)_, *I*_K(M)_ was relatively resistant to alteration by the presence of RIB. As depicted in Figure 6A, RIB’s presence noticeably decreased *I*_K(M)_ amplitudes in response to 1-s membrane depolarization from −50 to −10 mV. For example, as cells were exposed to 10 μM RIB, the *I*_K(M)_ amplitude measured at the end of the depolarizing step appreciably decreased from 76.3 ± 8 to 41.2 ± 6 pA (n = 7, *p* < 0.05). Figure 6B illustrates that RIB concentration-dependently depresses the amplitude of *I*_K(M)_ activated during step depolarization, with an estimated IC_50_ of 13.3 μM, a value that is relatively higher than that needed for the inhibition of *I*_K(erg)_ mentioned above. Under our experimental conditions, the amplitude of *I*_K(M)_ in the presence of RIB was compared with that after the subsequent addition of XE991 (10 μM). XE991 is reportedly a selective inhibitor of *I*_K(M)_ [35].

### 3.7. Lack of RIB Effect on the Perturbation of Hyperpolarization-Activated Cation Current (I_h_) in GH_3_ Cells

In the following experiments, we further studied whether RIB’s presence was able to perturb another type of ionic current, *I*_h_. Cells were exposed to Ca^2+^-free, Tyrode’s solution containing 1 μM tetrodotoxin, and the electrode was filled with a K^+^-containing solution. As shown in Figure 7A,B, when the 2-s hyperpolarizing voltage command changed from −40 to −110 mV, *I*_h_ with a slowly activating property was robustly evoked [46,47,48]. However, 1 min of cell exposure to RIB at a concentration of 10 μM did not affect the magnitude or gating (i.e., kinetic properties of current activation or deactivation) of *I*_h_ in response to long-lasting membrane hyperpolarization. For example, at −110 mV, the current amplitudes taken at the end of hyperpolarizing step obtained before and during the exposure to μM RIB were not significantly different (78 ± 5 pA [control] versus 77 ± 6 pA [in the presence of RIB]; n = 8, *p* > 0.05). However, with continued maintenance of RIB (10 μM), the subsequent addition of cilobradine (10 μM) or lamotrigine (10 μM) effectively inhibited or increased the *I*_h_ amplitude, respectively (Figure 7B). Cilobradine was recently reported to decrease *I*_h_ amplitude in pituitary cells [47], while lamotrigine could activate *I*_h_ in cortical neurons [49]. As such, distinct from its perturbing actions on different types of whole-cell K^+^ currents described above, RIB’s addition failed to perturb the magnitude, or gating kinetics of *I*_h_ identified in GH_3_ cells.

### 3.8. Effect of RIB on I_K(erg)_ Identified in HL-1 Atrial Cardiomyocytes

The administration of RIB reportedly prolongs QTc and induces cardiac arrhythmias [14,15]. In the final set of experiments, we wanted to test whether the presence of RIB could have any influence on the *I*_K(erg)_ present in heart cells (e.g., HL-1 heart cells) [32]. The measurements were conducted in these cells bathed in high-K^+^, Ca^2+^-free solution containing 1 μM tetrodotoxin and 0.5 mM CdCl_2_, and the pipette was filled with K^+^-containing solution. The examined cell was clamped at −10 mV, and various hyperpolarizing steps were delivered to evoke *I*_K(erg)_ [32]). As illustrated in Figure 8, when cells were exposed to 10 μM RIB, the *I–V* relationship of *I*_K(erg)_ decreased in membrane potentials ranging between 110 and −20 mV. For example, at the level of −100 mV, the addition of 10 μM RIB decreased the amplitude of deactivating *I*_K(erg)_ from 582 ± 42 to 205 ± 28 pA (n = 7, *p* < 0.05); and, the whole-cell conductance of *I*_K(erg)_ measured at the voltages ranging between −90 and −60 mV was decreased to 3.6 ± 0.7 nS (n = 7, *p* < 0.05) from a control value of 7.5 ± 1.1 nS (n = 7). After the compound’s washout, the current amplitude returned to 562 ± 38 pA (n = 7, *p* < 0.05). Therefore, similar to the observations described above in pituitary GH_3_ cells, *I*_K(erg)_ is functionally active in HL-1 cells and is sensitive to RIB’s presence.

## 4. Discussion

In this study, cell exposure to RIB decreased *I*_K(erg)_ amplitude and elevated the deactivation time-course of the current. Concentration-dependent inhibition of *I*_K(erg),_ with an effective IC_50_ of 2.7 μM, was observed in the presence of RIB. The addition of this compound also reduced the magnitude of voltage-dependent hysteresis for *I*_K(erg)_ in response to long-lasting triangular ramp pulses. Moreover, sustained exposure resulted in differential inhibition of peak or sustained *I*_K(DR)_ with an estimated IC_50_ of 28.7 M or 11.4 μM, respectively, combined with a pronounced slowing in current inactivation responding to long-maintained depolarization. RIB-mediated effectiveness in the modifications of ionic currents stated above could be a confounding factor affecting neuronal function. Simultaneously, they are independent of changes in synaptic efficacy [1,45] since the compound can readily cross the blood–brain barrier, thereby leading to a retardation in the aberrant growth of brain metastasis [50,51,52]. The modulatory (e.g., antineoplastic) actions of RIB on electrically excitable cells (e.g., neurons and endocrine or neuroendocrine cells) are not only mediated necessarily through decreased CDK4/6 activity but also via direct perturbations of the strength or gating kinetics in *I*_K(erg)_ and *I*_K(DR)_. In the graphical abstract, we illustrated the possible mechanism of RIB-induced perturbations in pituitary tumor and atrial cells.

According to studies on RIB pharmacokinetics after oral administration of 20 mg/kg, the peak plasma concentration was approximately 1036 ng/mL (i.e., about 2.38 μM) [50,53]. Therefore, the concentration of RIB required for the inhibition of ionic currents in GH_3_ and HL-1 cells appears to be higher than that used to suppress the activity of CDK4/6. However, after oral dosing with RIB, the highest tissue concentrations of RIB were found in tumor tissues, including pituitary tumors [9,51,53]. Of note, since the sensitivity of excitable cells to RIB could rely on the firing pattern of action potentials, pre-existing membrane potential, cytosolic Ca^+^ level, and the RIB concentrations used, the concentration of RIB required for modifying both the *I*_K(erg)_ magnitude and gating kinetics demonstrated here is achievable and, thus, of pharmacological relevance in humans. Indeed, oral administration of RIB produces perturbations in cardiac repolarization manifested by the QTc interval [14,15]. However, the extent to which RIB-induced venous thromboembolism is intimately linked to its effects on ionic current also remains to be further investigated [54].

There seems to be a discrepancy in the IC_50_ values between RIB-mediated inhibition of K^+^ currents (i.e., around 3 μM) and its effect on CDK inhibition (i.e., approximately 10 nM). The explanation for this difference is currently unknown; however, it could be due to the different experimental maneuvers used in the study involved. For investigations on CDK inhibition, surface-membrane components in host cells could have mostly been lysed and removed; consequently, the RIB molecules are likely to readily reach cytosolic enzymes (i.e., CDK-4/6) [11,12,13]. Conversely, in our experimental conditions, intact cells were required to study different types of membrane ionic currents. However, before RIB enters the cell interior, it certainly needs to contact with the surface membrane. Based on the considerations stated here, it seems likely that RIB’s effect on ionic current is of therapeutic relevance [14,15].

The modifications by RIB of membrane ionic currents demonstrated herein tend to be acute in onset. Hence, the actions could be independent of its interaction with the activity of cytosolic CDK-4/6, as stated previously [2,5,14,55]. The action of this drug on the magnitude or gating of *I*_K(erg)_ is thought to occur either through its preferential binding to the open state of the *KCNH* channel or through interaction with the channel’s open conformation.

The phenomenon of the voltage-dependent hysteresis of ionic currents has been proposed to play an essential role in affecting the electrical behavior of excitable cells [42]. In the present observations, consistent with previous work on HCN or KCNQ channels [42,43,56], the K_erg_ channels present in GH_3_ cells underwent a hysteresis in which there is likely to be a mode shift because the voltage sensitivity of the gating change movement relies on the previous state of the channel [42]. Moreover, we further explored the possible perturbation by RIB of the non-equilibrium property inherent to K_erg_ channels. These results highlight RIB’s ability to lessen the Δarea of hysteresis for *I*_K(erg)_ elicited in response to long-lasting triangular ramp pulses. In addition to the decreased current amplitudes, RIB’s presence could produce a slowing in the deactivating time-course of *I*_K(erg)_ responding to sustained hyperpolarization. Therefore, any perturbations of *I*_K(erg)_ induced by RIB depend not only on the RIB concentration used but also on different factors that include the pre-existing resting potential or varied firing patterns of action potentials.

In this study, subsequent addition of 17β-estradiol, PMA, or transforming growth factor-β, still in the presence of RIB, did not modify the *I*_K(erg)_ elicited in response to membrane hyperpolarization in GH_3_ cells. These compounds were previously demonstrated to induce the induction of the CDK-4/6 complex [39,40,41]. Thus, we proposed that the ability of RIB to perturb the amplitude or gating of *I*_K(erg)_ in GH_3_ or HL-1 cells appears to be direct and hence would not be associated with its inhibition of cytosolic CDK-4/6 activity. However, these cells might broadly express the activity of the CDK complex [28,29]. Thus, caution must be observed in attributing the action (e.g., clinical efficacy) of RIB on either neuronal or cardiac function avidly to the inhibition of CDK-4/6.

The inhibitory action of RIB on *I*_K(DR)_ demonstrated in GH_3_ cells was observed to correlate over time with an apparent increase in the inactivation rate of the current elicited during depolarizing voltage commands with long durations (i.e., 1 or 10 s). The IC_50_ value required for RIB-induced inhibition of peak or sustained *I*_K(DR)_ responding to 1-s membrane depolarization was estimated to be 28.7 or 11.4 μM, respectively, while that for its inhibitory effect on *I*_K(M)_ was 13.3 μM. The blocking site of RIB could be located within the K_V_ channel pore in situations where the channel remains mostly open. That is, the RIB molecule is allowed to reach the blocking site only when the channel is in the open state. It likely exerts an inhibitory effect on *I*_K(DR)_ through C-type inactivation involving conformational changes at the external mouth of the channel pore [30]. However, a lack of impact of RIB on hyperpolarization-evoked *I*_h_ was observed in GH_3_ cells. At any rate, inhibition of *I*_K(erg)_ and *I*_K(DR)_ produced by this agent’s presence might potentially modify the electrical behaviors of excitable cells. The concerted effectiveness of RIB in the suppression of CDK-4/6 activity, as well as direct inhibition of IK(erg) and IK(DR), may synergistically affect the functional activities of neoplastic or heart cells occurring in vivo.

It is worth noting that CDK 4/6 inhibitors have changed the therapeutic approach for hormone receptor-positive metastatic breast cancer [5,57]. However, the pharmacological or toxicological properties have been reported to be quite different among RIB and structurally similar compounds [5,15,55]. Thus, it is quite tempting to anticipate that some of the reasons for these discrepancies result from the possible perturbations on different types of ionic currents shown here. Changes in Kerg channels’ activity have been reported to affect the proliferation or apoptosis of neoplastic cells [24,25,26], and these channels are broadly expressed in different types of neoplastic cells, including breast cancer cells [24,58]. Nonetheless, perturbations in the magnitude or gating of *I*_K(erg)_ and *I*_K(DR)_ produced by the presence of RIB may potentially converge to act on the functional activities of neurons, endocrine or neuroendocrine cells, and cardiac myocytes (e.g., GH_3_ and HL-1 cells) [3,6,13,14].

## Figures and Tables

**Figure 1 ijms-21-08078-f001:**
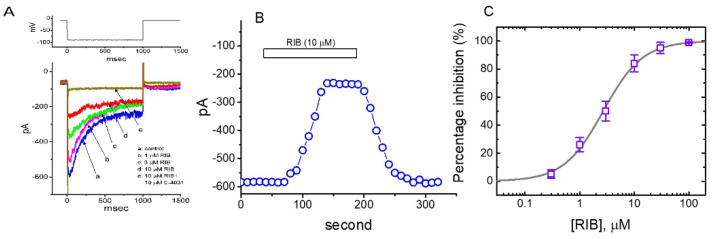
Effect of RIB on *erg*-mediated K^+^ current (*I*_K(erg)_) recorded from voltage-clamped pituitary GH_3_ cells. In these experiments, we kept cells to be bathed in high-K^+^, Ca^2+^-free solution and the recording microelectrode was filled up with K^+^-containing solution. (**A**) Representative *I*_K(erg)_ traces achieved in the absence (a) and presence of 1 μM RIB (b), 3 μM RIB (c), 10 μM RIB (d), or 10 μM RIB plus 10 μM E-4031 (e). The upper part shows the voltage profile applied. (**B**) Time course of RIB (10 μM)-mediated change on *I*_K(erg)_ amplitude. Current amplitude was measured at the beginning of hyperpolarizing pulse from −10 to −90 mV with a duration of 1 s at a rate of 0.1 Hz. The horizontal bar denotes the application of RIB (10 μM). (**C**) Concentration-dependent inhibition of RIB on *I*_K(erg)_ (i.e., E-4031-sensitive current) responding to 1-s membrane hyperpolarization (mean ± SEM; n = 8 for each point). Current amplitude was taken at the end of 1-s hyperpolarizing pulse applied from −10 to −90 mV. The modified Hill equation elaborated under Materials and Methods was well fit to the experimental observations (continuous solid line).

**Figure 2 ijms-21-08078-f002:**
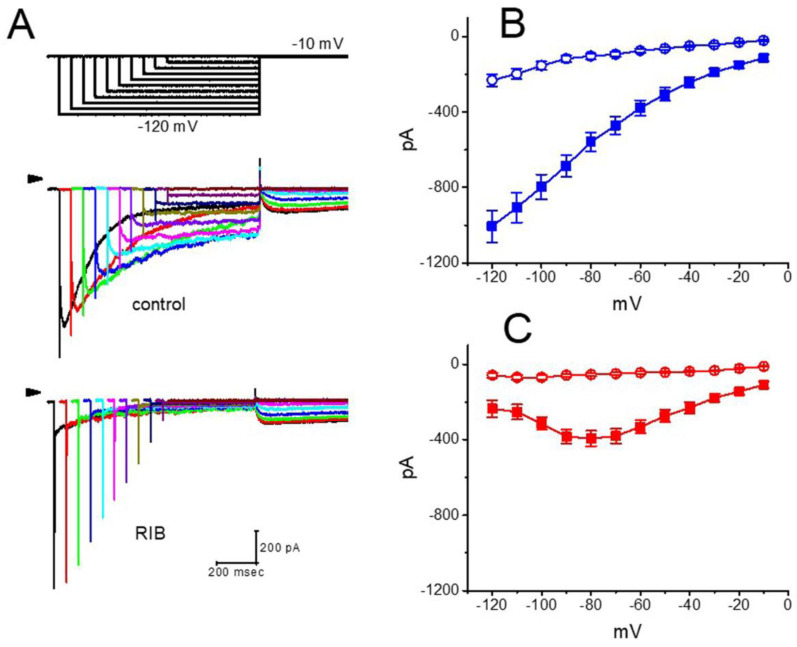
Effect of RIB on mean current-voltage (*I–V*) relationships of *I*_K(erg)_ identified in GH_3_ cells. In (**A**), the uppermost graph depicts the voltage profile applied, and below are representative *I*_K(erg)_ traces obtained in the absence (upper) and presence of 10 μM RIB (lower). The horizontal arrow in the left side of current traces indicates the zero-current level, and the calibration mark indicated in the right lower corner applies for all traces. In (**B**,**C**), mean *I–V* relationships of *I*_K(erg)_ amplitude were respectively measured at the beginning (blue symbols) and end (red symbols) of various hyperpolarizing pulses achieved in the absence (filled square symbols) or presence (open circle symbols) of 10 μM RIB (mean ± SEM; n = 8 for each point).

**Figure 3 ijms-21-08078-f003:**
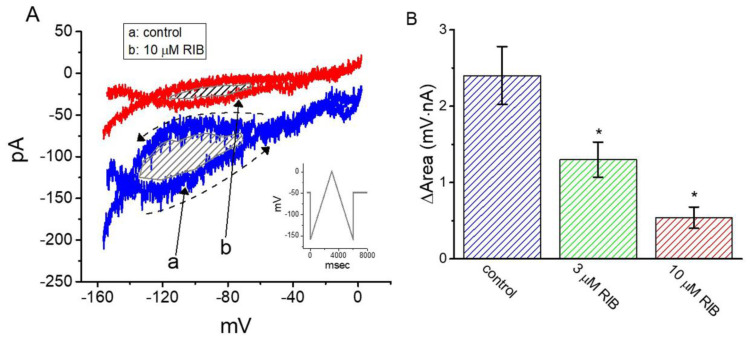
Effect of RIB on the voltage hysteresis of *I*_K(erg)_ identified in GH_3_ cells. (**A**) Representative current traces elicited in response to 6-s long-duration triangular (i.e., upsloping [forward] and downsloping [reverse]) ramp pulse between −160 and 0 mV. Inset in (**A**) shows the voltage profile delivered during the measurements. Current trace labeled a is control (i.e., the absence of RIB), and that labeled b was taken during the exposure to 10 μM RIB. Arrows indicate the direction of *I*_K(erg)_ trajectory in which time passes. (**B**) Summary bar graph showing the effect of RIB on the area (as indicated in the shaded area of (**A**)) of voltage hysteresis (mean ± SEM; n = 8 for each bar). * Significantly different from control (*p* < 0.05).

**Figure 4 ijms-21-08078-f004:**
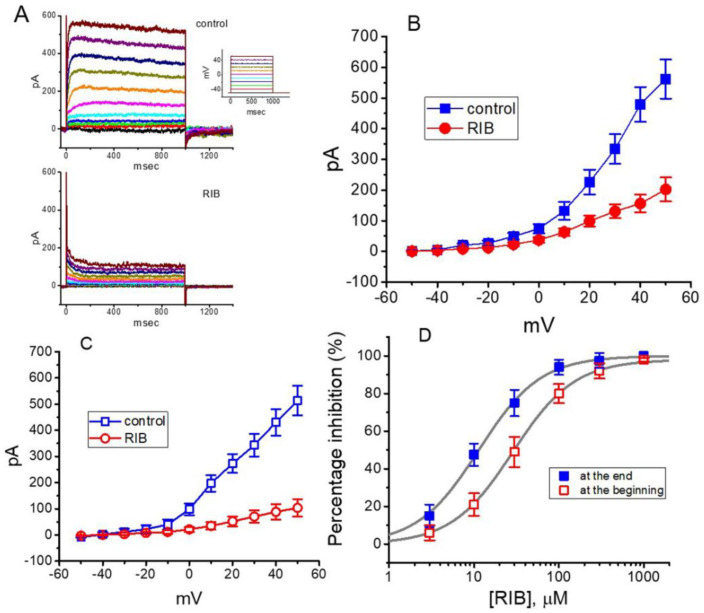
Effect of RIB on delayed-rectifier K^+^ current (*I*_K(DR)_) identified in GH_3_ cells. Cells were kept bathed in Ca^2+^-free, Tyrode’s solution containing 1 μM tetrodotoxin, and the recording electrode was filled with K^+^-containing solution. (**A**) Representative *I*_K(DR)_ traces obtained in the absence (upper) or presence (lower) of 10 μM RIB. In this set of experiments, we voltage-clamped cells at −50 mV and then delivered a series of the voltages ranging between −50 and +50 mV in a duration of 1 s (as indicated in Inset). In (**B**,**C**), mean *I–V* relationships of *I*_K(DR)_ respectively obtained at the beginning and end of the command voltages were taken during the control (square symbols) or after the addition of 10 μM RIB (circle symbols) (mean ± SEM; n = 8 for each point). The *I*_K(DR)_ amplitude was measured at the beginning (**B**), filled symbols) or end (**C**), open symbols) of 1-s depolarizing pulse from −50 mV to different command steps. In (**D**), the relations between the percentage inhibition of the peak (□) or sustained component (■) of *I*_K(DR)_ (i.e., TEA-sensitive current) in response to 1-s depolarizing pulse are illustrated. Current amplitudes taken at the beginning or end of depolarizing steps from −50 to +50 mV in the absence or presence of different RIB concentrations were compared with the control values. The continuous lines overlaid were appropriately fitted by a modified Hill function (see text for details). The IC_50_ value needed for the inhibition of the peak (□) or sustained (■) *I*_K(DR)_ in the presence of RIB was appropriately estimated; and, the plots yielded IC_50_ value to be 28.7 or 11.4 μM, respectively.

**Figure 5 ijms-21-08078-f005:**
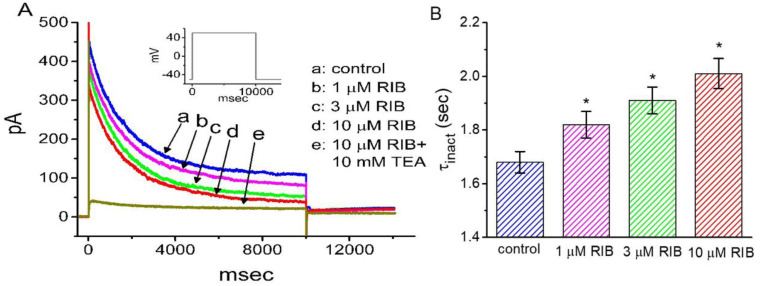
Effect of RIB on *I*_K(DR)_ amplitude in response to 10-s long depolarizing step. (**A**) Representative *I*_K(DR)_ elicited by 10-s depolarizing pulse from −50 to +50 mV (indicated in the Inset). a: control; b: 1 μM RIB; c: 3 μM RIB; d: 10 μM RIB; e: 10 μM RIB plus 10 mM TEA. (**B**) Summary bar graph showing the effect of RIB on the τ_inact_ value of long hyperpolarization-induced *I*_K(DR)_ (mean ± SEM; n = 7 for each bar). The *I*_K(DR)_ was readily evoked by 10-s depolarizing command step from −50 to +50 mV. * Significantly different from control (*p* < 0.05).

**Figure 6 ijms-21-08078-f006:**
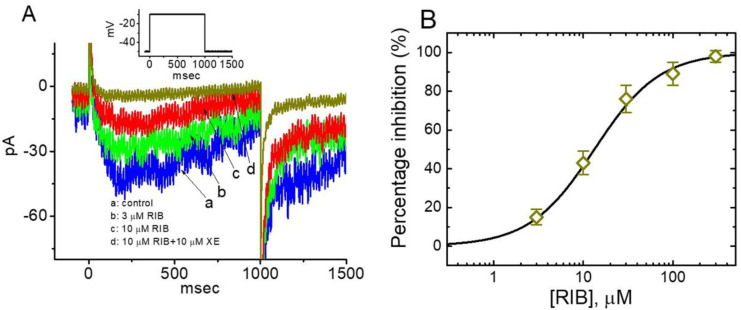
Effect of RIB on M-type K^+^ current (*I*_K(M)_) in GH_3_ cells. The experiments were conducted in cells bathed in high-K^+^, Ca^2+^-free solution and the recording electrode was filled up in K^+^-containing internal solution. (**A**) Representative current traces in response to 1-s membrane depolarization from −50 to −10 mV. Inset shows the voltage protocol applied. XE: XE991. (**B**) Concentration-dependent inhibition of *I*_K(M)_ (i.e., XE991-sensitive current) produced by RIB. Each cell was voltage-clamped at −50 mV, and the current amplitude obtained during cell exposure to different RIB concentration was thereafter measured at the end of 1-s depolarization to −10 mV. The value for IC_50_ of RIB-induced inhibition of XE991-sensitive current or for Hill coefficient was estimated to be 13.3 μM or 1.2, respectively. Continuous sigmoidal line, on which the data points were overlaid, represents best fit to the modified Hill equation.

**Figure 7 ijms-21-08078-f007:**
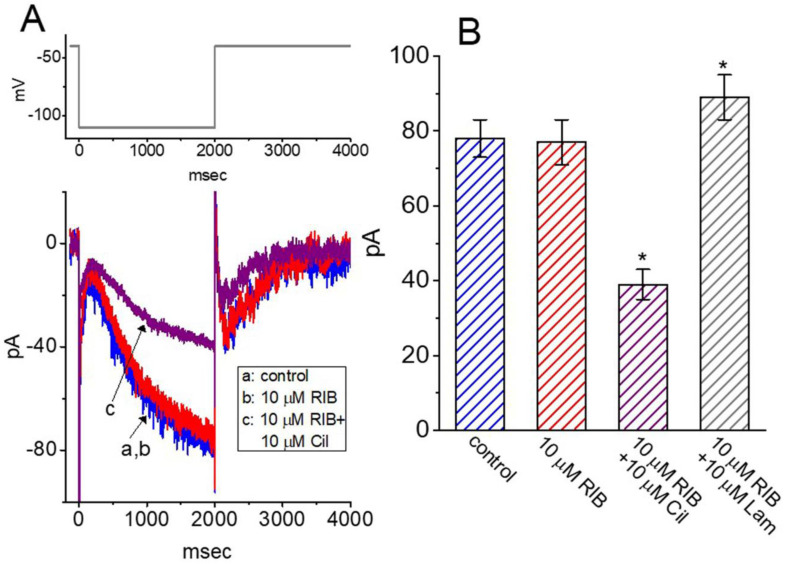
Failure of RIB to perturb hyperpolarization-activated cation current (*I*_h_) in GH_3_ cells. In these recordings, cells were bathed in Ca^2+^-free, Tyrode’s solution and we filled up the electrode by using K^+^-containing solution. (**A**) Representative *I*_h_ traces obtained in the control (a) and in the presence of 10 μM RIB (b) or 10 μM RIB plus 10 μM cilobradine (Cil) (c). The upper part shows the voltage-clamp protocol applied. (**B**) Summary bar graph depicting the effect of RIB, RIB plus cilobradine (Cil), and RIB plus lamotrigine (Lam) on the *I*_h_ amplitude responding to hyperpolarizing pulse (mean ± SEM; n = 8 for each bar). Each cell was 2-s hyperpolarized from −40 to −110 mV, and the current amplitude was taken at the end of the hyperpolarizing command step. * Significantly different from control group or 10 μM RIB alone group (*p* < 0.05).

**Figure 8 ijms-21-08078-f008:**
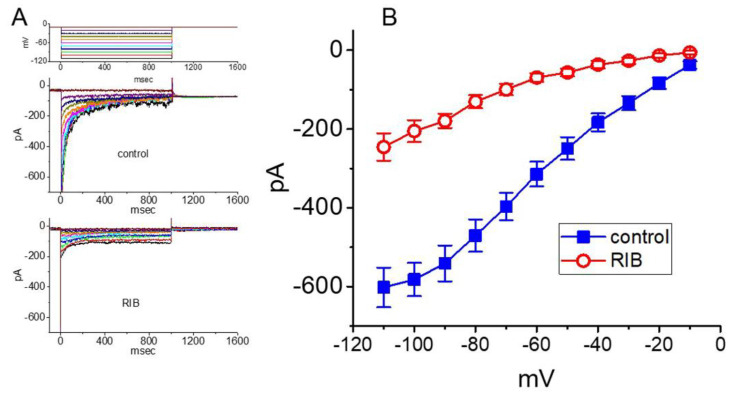
Inhibitory effect of RIB on *I*_K(erg)_ recorded from HL-1 atrial cardiomyocytes. Cells were immersed in high-K^+^, Ca^2+^-free solution containing 1 μM tetrodotoxin and 0.5 mM CdCl_2_. (**A**) Representative current traces obtained in response to different hyperpolarizing steps from a holding potential of −10 mV (as indicated in the uppermost part). Current traces shown in the upper or lower part indicate the absence or presence of 10 μM RIB, respectively. (**B**) Mean *I–V* relationship of *I*_K(erg)_ obtained in control (■) and during exposure to 10 μM RIB (○). The results were obtained at the beginning of voltage pulses (mean ± SEM; n = 7 for each point).

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
