# Peer review of "Characterization of the Synergistic Inhibition of IK(erg) and IK(DR) by Ribociclib, a Cyclin-Dependent Kinase 4/6 Inhibitor"

_ijms, 2020, doi:10.3390/ijms21218078_

Round 1
Reviewer 1 Report
The authors have addressed my concerns on this manuscript and I have no further comment. Now it should be ready for publication.
Author Response
Dear Reviewer,
We appreciate all your comments which make this work more comprehensive and complete.
Reviewer 2 Report
In their revised manuscript entitled “Characterization in synergistic effective inhibition of 2 IK(erg) and IK(DR) by ribociclib, an CDK-4/6 inhibitor”, Pin-Yen Liu et al hav tried to take into consideration the points that have been raised with respect to the first version as described below. The replies of this reviewer to the responses of the author are inserted at the end of each of these points.
Point 1) Currents shown in figure 6 cannot be M currents, as these currents activate slowly and do not show any inactivation during 1 s depolarizations. To verify that these currents were mediated by M channels (i.e. Kv7 channels), the authors would need to use appropriate blockers (e.g. XE991) and to apply ribociclib in the absence and presence of the chosen blocker.
Author reply 1. Thanks for the comments pointed out by the reviewer. As advised by the reviewer, an additional set of experiments regarding the measurements studied by adding appropriate blockers (e.g., XE991) were performed. The experimental results were hence included in the revised version of the manuscript. We believe that, consistent with previous observations (Sankaranarayanan and Simasko, J Neurosci 1996;16:1668; Selyanko et al., J Neurosci 1999;19:7742), currents shown in Figure 6 could be M-type K+ currents. Additionally, the text appearing in the revised manuscript was correspondingly revised (indicated in the red color).
Reviewer reply 1: On p. 4, line 154 one can read: The IK(erg), IK(DR), or IK(M) 155 amplitudes during the application of RIB were compared with those compared with those measured 156 after subsequent addition of E-4031 (10 M), tetraethylammonium chloride (TEA; 10 mM), or XE991 157 (10 M), respectively. E-4031 or XE991 is known to be a selective blocker of IK(erg) or IK(M), respectively 158 [21, 35], and TEA can effectively suppress the amplitude of IK(DR).
Further down (p. 4, line 166), one can read: …[RIB] is the ribociclib concentration; and, Emax indicates the maximal 166 inhibition of IK(erg) (i.e., E-4031-sensitive current), IK(DR) (i.e., TEA-sensitive current), or IK(M) (i.e., XE991-167 sensitive current) amplitude produced by the presence of RIB.
With the exception of figures 6 and 7, the authors fail to show currents determined in the presence of any of these blockers. Moreover, it remains entirely unclear how inhibition of E-4031/TEA/XE991-sensitive currents by ribociclib has been determined. This needs to be amended! Original current traces obtained in the presence of blockers should be shown!
Point 2) The same holds true for the other types of currents reported in the manuscript. For each current type, ribociclib needs to be tested in the absence and presence of selective blockers to prove specificity of inhibitory actions.
Author reply 2. As per the suggestion pointed out by the reviewers, the specific blocker used for each ionic current was tested and the text in the revised manuscript (indicated in the red color) was hence appropriately revised or modified.
Reviewer reply 2: see reviewer reply 1.
Point 3) In the discussion, one can read the following statement (p. 13, line 417): “Therefore, the concentration of RIB required for the inhibition of ionic currents present in GH3 and HL-1 cells appears to be mildly higher than that used to suppress the activity of CDK4/6.” Given the fact that IC50 values for CDK inhibition are in the range of 0.01 µM and IC50 values within the manuscript are 2.7 µM, this is hard to reconcile.
Author reply 3. According to previous studies RIB pharmacokinetics, as the compound was orally administered at 20 mg/kg, the peak plasma concentration was reported to reach about 1036 ng/ml (i.e. about 2.38 µM). Therefore, the concentration of RIB required for the inhibition of ionic currents present in GH3 and HL-1 cells appears to be slightly higher than that used to suppress the activity of CDK4/6. Of further notice, in addition to the inhibition of current amplitude, the presence of RIB could modify the voltage-dependent hysteresis and the gating of IK(erg). Therefore, the sensitivity of excitable cells to RIB could rely on the firing pattern of action potentials, preexisting membrane potential, and the RIB concentrations used. By extension, after oral dosing with RIB, the highest tissue concentrations of RIB were noticed in tumor tissues including pituitary tumor (Miller et al., 2019; Nguyen et al., 2019; Lamb et al., 2020). Hence, it is likely that the concentration of RIB used for the perturbations of ionic currents would be clinically achievable.
Reviewer reply 3: My critique was not at all related to pharmacokinetics and independent thereof. Given that IC50 values for CDK inhibition are 10 nM and those for current inhibition are 3000 nM (factor 300!!!), on cannot argue that “, the concentration of RIB required for the inhibition of ionic currents present in GH3 and HL-1 cells appears to be mildly higher than that used to suppress the activity of CDK4/6”. This reasoning must be changed!
Point 4) The authors claim that (p. 13, line416) “plasma concentration was currently reported to reach about 1036 ng/ml (i.e., about 2.38 µM)”. In this context, they forget to mention plasma protein binding of ribociclib in the range of 70 to 90 % (Bao et al, J Pharm Biomed Anal. 2019 Mar 20;166:197-204). Hence, free and active concentrations would be considerably lower. Moreover, most pharmacokinetic studies of ribociclib report plasma concentrations below 1 µM.
Author Reply 4. Thanks for the specific comments raised by the reviewer. According to the studies on RIB pharmacokinetics, as the compound was orally administered at 20 mg/kg, the peak plasma concentration was currently reported to reach about 1036 ng/ml (i.e. about 2.38 µM) (Martínez-Chávez et al., 2019; Miller et al., 2019). Moreover, a recent report showed that plasma protein binding of RIB was in the range of 70-90% (Bao et al., 2019). Therefore, free and active concentrations of this compound tend to be lower. Therefore, the concentration of RIB required for the inhibition of ionic currents present in GH3 and HL-1 cells appears to be higher than that used to suppress the activity of CDK4/6. However, after oral dosing with RIB, the highest tissue concentrations of RIB were found in tumor tissues including pituitary tumor (Miller et al., 2019; Nguyen et al., 2019; Lamb et al., 2020). Of notice, since the sensitivity of excitable cells to RIB could rely on the firing pattern of action potentials, preexisting membrane potential, and the RIB concentrations achieved, the concentration of RIB used for its modifications on both IK(erg) magnitude and kinetics demonstrated here is achievable and, thus, of clinical relevance in humans. Indeed, oral administration of RIB was recently noticeably demonstrated to cause the possible perturbations on cardiac repolarization manifested by QTc interval (Barber et al., 2019; Rascon et al., 2019; Santoni et al., 2019). This issue was included in the Discussion section of the revised manuscript, and an additional paper was included in Reference section of the revised manuscript.
Reviewer reply 4: Given that peak plasma concentrations reach 2.38 µM and plasma protein binding is at least 70%, free ribociclib plasma concentrations cannot surmount 700 nM. According to figure 1, this concentration does not cause significant current inhibition. With respect to concentrations of RIB in tumor tissues, it remains unclear which concentrations might be sensed by ion channels. Hence, the authors must be more careful regarding the clinical relevance of their findings. In the current version, it is pharmacologically implausible to follow their respective argumentation.
Point 5) The text is hard to understand because of problems regarding the English language; a few examples:
- 2, line 47: “CDK inhibitors (e.g., RIB) are thought to be known targets to become deregulated in 4different types of tumor cells including pituitary tumor cells”: how can a CDK inhibitor be a target and become deregulated?
- line 50: “CDK-4/6 inhibitors (e.g., RIB) have also been reported to protect the degenerative changes of motor neurons”: when degenerative changes are protected, what should be the therapeutic benefit?
- 8, line 273: “In the next stage of experiments, we wanted to examine assess if this drug has any perturbations on IK(DR) inherently in these cells”: what do the authors mean by examine assess?
Author reply 5. (1) Thanks for the advice by the reviewer. The sentence was appropriately rephrased to “The CDK inhibitors (e.g., RIB) are thought to be known targets, by which different types of tumor cells including pituitary tumor cells become deregulated”.
(2) Based on the experimental results from previous study by Hor et al., 2018, in an organoid model of spinal muscular atrophy, the degenerative change in motor neurons could be progressively deteriorated. However, as model was exposed to CDK-4/6 inhibitors (e.g., RIB), degeneration in those neurons could be noticed to be prevented. The results suggest that the presence of RIB might protect the degenerative change in motor neurons in that model. However, of importance, how the underlying ionic mechanism of this action remains unknown.
(3) Thanks for the reviewer’s comment. Goof!, We made a mistake. “assess” was removed from this sentence in the revised manuscript.
Reviewer reply 5: As reviewer, I don’t feel like being in charge for the quality of the entire text. In the first report, some examples were presented in order to document that the text is full of spelling, grammatical and syntax errors. Unfortunately, this hasn’t changed. Therefore, the text should be corrected by a knowledgeable native speaker prior to resubmission.
Author Response
Authors’ responses for the manuscript ijms-962051
Title: Characterization of the synergistic inhibition of IK(erg) and IK(DR)
by ribociclib, a cyclin-dependent kinase 4/6 inhibitor
Dear editors and reviewers,
We appreciate all the comments from reviewers and editors on this paper. We have revised the manuscript accordingly. Hopefully, we have adequately responded to all critiques and satisfied the reviewers and editors.
Point 1) Currents shown in figure 6 cannot be M currents, as these currents activate slowly and do not show any inactivation during 1 s depolarizations. To verify that these currents were mediated by M channels (i.e. Kv7 channels), the authors would need to use appropriate blockers (e.g. XE991) and to apply ribociclib in the absence and presence of the chosen blocker.
Author reply 1. Thanks for the comments pointed out by the reviewer. As advised by the reviewer, an additional set of experiments regarding the measurements studied by adding appropriate blockers (e.g., XE991) were performed. The experimental results were hence included in the revised version of the manuscript. We believe that, consistent with previous observations (Sankaranarayanan and Simasko, J Neurosci 1996;16:1668; Selyanko et al., J Neurosci 1999;19:7742), currents shown in Figure 6 could be M-type K+ currents. Additionally, the text appearing in the revised manuscript was correspondingly revised (indicated in the red color).
Reviewer reply 1: On p. 4, line 154 one can read: The IK(erg), IK(DR), or IK(M) 155 amplitudes during the application of RIB were compared with those compared with those measured 156 after subsequent addition of E-4031 (10 M), tetraethylammonium chloride (TEA; 10 mM), or XE991 157 (10 M), respectively. E-4031 or XE991 is known to be a selective blocker of IK(erg) or IK(M), respectively 158 [21, 35], and TEA can effectively suppress the amplitude of IK(DR).
Further down (p. 4, line 166), one can read: …[RIB] is the ribociclib concentration; and, Emax indicates the maximal 166 inhibition of IK(erg) (i.e., E-4031-sensitive current), IK(DR) (i.e., TEA-sensitive current), or IK(M) (i.e., XE991-167 sensitive current) amplitude produced by the presence of RIB.
With the exception of figures 6 and 7, the authors fail to show currents determined in the presence of any of these blockers. Moreover, it remains entirely unclear how inhibition of E-4031/TEA/XE991-sensitive currents by ribociclib has been determined. This needs to be amended! Original current traces obtained in the presence of blockers should be shown!
Response to Reviewer reply 1:
We appreciate your comment and Figures 1A and 5A were redone for the perusal, since additional current traces (i.e., RIB plus E-4031 or RIB plus TEA) were included in each panel.
Point 2) The same holds true for the other types of currents reported in the manuscript. For each current type, ribociclib needs to be tested in the absence and presence of selective blockers to prove specificity of inhibitory actions.
Author reply 2. As per the suggestion pointed out by the reviewers, the specific blocker used for each ionic current was tested and the text in the revised manuscript (indicated in the red color) was hence appropriately revised or modified.
Reviewer reply 2: see reviewer reply 1.
Response to Reviewer reply 2: We appreciate your comment and Figures 1A and 5A were redone for the perusal, since additional current traces (i.e., RIB plus E-4031 or RIB plus TEA) were included in each panel.
Point 3) In the discussion, one can read the following statement (p. 13, line 417): “Therefore, the concentration of RIB required for the inhibition of ionic currents present in GH3 and HL-1 cells appears to be mildly higher than that used to suppress the activity of CDK4/6.” Given the fact that IC50 values for CDK inhibition are in the range of 0.01 µM and IC50 values within the manuscript are 2.7 µM, this is hard to reconcile.
Author reply 3. According to previous studies RIB pharmacokinetics, as the compound was orally administered at 20 mg/kg, the peak plasma concentration was reported to reach about 1036 ng/ml (i.e. about 2.38 µM). Therefore, the concentration of RIB required for the inhibition of ionic currents present in GH3 and HL-1 cells appears to be slightly higher than that used to suppress the activity of CDK4/6. Of further notice, in addition to the inhibition of current amplitude, the presence of RIB could modify the voltage-dependent hysteresis and the gating of IK(erg). Therefore, the sensitivity of excitable cells to RIB could rely on the firing pattern of action potentials, preexisting membrane potential, and the RIB concentrations used. By extension, after oral dosing with RIB, the highest tissue concentrations of RIB were noticed in tumor tissues including pituitary tumor (Miller et al., 2019; Nguyen et al., 2019; Lamb et al., 2020). Hence, it is likely that the concentration of RIB used for the perturbations of ionic currents would be clinically achievable.
Reviewer reply 3: My critique was not at all related to pharmacokinetics and independent thereof. Given that IC50 values for CDK inhibition are 10 nM and those for current inhibition are 3000 nM (factor 300!!!), on cannot argue that “, the concentration of RIB required for the inhibition of ionic currents present in GH3 and HL-1 cells appears to be mildly higher than that used to suppress the activity of CDK4/6”. This reasoning must be changed!
Response to Reviewer reply 3:
As pointed out by the reviewer, there seems to be a discrepancy in IC50 value between the inhibition of K+ currents (i.e., around 3 M) and its effect on CDK inhibition (i.e., around 10 nM). The explanation in such difference is currently unknown. However, it could be due to various experimental maneuvers used in the study applied. For investigations on CDK inhibition, surface-membrane components in host cells could have mostly been lysed and removed; consequently, the RIB molecules is likely to readily reach cytosolic enzymes (Liu et al., 2002; Musat et al., 2004; Aristizabal et al., 2018). Conversely, in our experimental conditions, intact cells were used to study different types of membrane ionic currents. However, before RIB enters the cell interior, it certainly needs to contact with the surface membrane. Based on the considerations stated here, it seems likely that RIB effect on ionic current is of therapeutic and clinical relevance (Barber et al., 2019; Santoni et al., 2019). The text relevant to this issue is important and was hence included in the Discussion section of the revised manuscript.
By the way, “mildly” could be inappropriate and hence was removed in the text of this version of the revised manuscript (line 1 from the bottom, page 19). The sentence has been revised as the following,
“Therefore, the concentration of RIB required for the inhibition of ionic currents present in GH3 and HL-1 cells appears to be mildly higher than that used to suppress the activity of CDK4/6.”
Point 4) The authors claim that (p. 13, line416) “plasma concentration was currently reported to reach about 1036 ng/ml (i.e., about 2.38 µM)”. In this context, they forget to mention plasma protein binding of ribociclib in the range of 70 to 90 % (Bao et al, J Pharm Biomed Anal. 2019 Mar 20;166:197-204). Hence, free and active concentrations would be considerably lower. Moreover, most pharmacokinetic studies of ribociclib report plasma concentrations below 1 µM.
Author Reply 4. Thanks for the specific comments raised by the reviewer. According to the studies on RIB pharmacokinetics, as the compound was orally administered at 20 mg/kg, the peak plasma concentration was currently reported to reach about 1036 ng/ml (i.e. about 2.38 µM) (Martínez-Chávez et al., 2019; Miller et al., 2019). Moreover, a recent report showed that plasma protein binding of RIB was in the range of 70-90% (Bao et al., 2019). Therefore, free and active concentrations of this compound tend to be lower. Therefore, the concentration of RIB required for the inhibition of ionic currents present in GH3 and HL-1 cells appears to be higher than that used to suppress the activity of CDK4/6. However, after oral dosing with RIB, the highest tissue concentrations of RIB were found in tumor tissues including pituitary tumor (Miller et al., 2019; Nguyen et al., 2019; Lamb et al., 2020). Of notice, since the sensitivity of excitable cells to RIB could rely on the firing pattern of action potentials, preexisting membrane potential, and the RIB concentrations achieved, the concentration of RIB used for its modifications on both IK(erg) magnitude and kinetics demonstrated here is achievable and, thus, of clinical relevance in humans. Indeed, oral administration of RIB was recently noticeably demonstrated to cause the possible perturbations on cardiac repolarization manifested by QTc interval (Barber et al., 2019; Rascon et al., 2019; Santoni et al., 2019). This issue was included in the Discussion section of the revised manuscript, and an additional paper was included in Reference section of the revised manuscript.
Reviewer reply 4: Given that peak plasma concentrations reach 2.38 µM and plasma protein binding is at least 70%, free ribociclib plasma concentrations cannot surmount 700 nM. According to figure 1, this concentration does not cause significant current inhibition. With respect to concentrations of RIB in tumor tissues, it remains unclear which concentrations might be sensed by ion channels. Hence, the authors must be more careful regarding the clinical relevance of their findings. In the current version, it is pharmacologically implausible to follow their respective argumentation.
Response to Reviewer reply 4:
As per the suggestion by the reviewer, the text in the revised version of this manuscript was appropriately rephrased. However, based on the current study, it needs to be emphasized that the IC50 value could not be single factor used for evaluation of its action on tumor tissues. Since the sensitivity of excitable cells to RIB could rely on various confounding factors, such as the firing pattern of action potentials, pre-existing membrane potential, the level of cytosolic Ca2+, and the RIB concentrations applied, the concentration of RIB used for its modifications on both IK(erg) magnitude and gating kinetics demonstrated here tends to be achievable and, thus, of clinical relevance in humans. More importantly, oral administration of RIB has noticeably demonstrated to cause the possible perturbations on cardiac repolarization manifested by QTc interval.
Point 5) The text is hard to understand because of problems regarding the English language; a few examples:
- 2, line 47: “CDK inhibitors (e.g., RIB) are thought to be known targets to become deregulated in 4different types of tumor cells including pituitary tumor cells”: how can a CDK inhibitor be a target and become deregulated?
- line 50: “CDK-4/6 inhibitors (e.g., RIB) have also been reported to protect the degenerative changes of motor neurons”: when degenerative changes are protected, what should be the therapeutic benefit?
- 8, line 273: “In the next stage of experiments, we wanted to examine assess if this drug has any perturbations on IK(DR) inherently in these cells”: what do the authors mean by examine assess?
Author reply 5. (1) Thanks for the advice by the reviewer. The sentence was appropriately rephrased to “The CDK inhibitors (e.g., RIB) are thought to be known targets, by which different types of tumor cells including pituitary tumor cells become deregulated”.
(2) Based on the experimental results from previous study by Hor et al., 2018, in an organoid model of spinal muscular atrophy, the degenerative change in motor neurons could be progressively deteriorated. However, as model was exposed to CDK-4/6 inhibitors (e.g., RIB), degeneration in those neurons could be noticed to be prevented. The results suggest that the presence of RIB might protect the degenerative change in motor neurons in that model. However, of importance, how the underlying ionic mechanism of this action remains unknown.
(3) Thanks for the reviewer’s comment. Goof!, We made a mistake. “assess” was removed from this sentence in the revised manuscript.
Reviewer reply 5: As reviewer, I don’t feel like being in charge for the quality of the entire text. In the first report, some examples were presented in order to document that the text is full of spelling, grammatical and syntax errors. Unfortunately, this hasn’t changed. Therefore, the text should be corrected by a knowledgeable native speaker prior to resubmission.
Response to Reviewer reply 5:
Thanks for the reviewer’s comments on ways to improve our manuscript. The text in this version of the revised manuscript has been appropriately modified. In fact, an additional paragraph was also incorporated into the Discussion section of the revised manuscript as the followings,
“There seems to be a discrepancy in IC50 value between RIB-mediated inhibition of K+ currents (i.e., around 3 M) and its effect on CDK inhibition (i.e., around 10 nM). The explanation in such difference is currently unknown; however, it could be due to different experimental maneuvers used in the study involved. For investigations on CDK inhibition, surface-membrane components in host cells could have mostly been lysed and removed; consequently, the RIB molecules is thus likely to readily reach cytosolic enzymes (i.e., CDK-4/6) [11-13]. Conversely, in our experimental conditions, intact cells were virtually needed to study different types of membrane ionic currents. However, before RIB enters the cell interior, it certainly needs to contact with surface membrane. Based on the considerations stated here, it seems likely that RIB effect on ionic current is of therapeutic relevance [14, 15].”
Round 2
Reviewer 2 Report
The authors have responded to the points of critique in a satisfactors manner. Only one minor issue remains to be resolved:
in Line 493 one can read: "Based on the considerations stated here, it seems likely that the RIB's effect on ionic current is of therapeutic relevance [14, 15]."
References 14 and 15 refer to cardiotoxic effects of RIB. Therefore, this sentence should be rephrased as follows: "Based on the considerations stated here, it seems likely that the RIB's effect on ionic current is of relevance with respect to cardiotoxicity [14, 15]."
This manuscript is a resubmission of an earlier submission. The following is a list of the peer review reports and author responses from that submission.
Round 1
Reviewer 1 Report
The manuscript by Pin-Yen Liu et al entitled “Characterization in synergistic effective inhibition of 2 IK(erg) and IK(DR) by ribociclib, an CDK-4/6 inhibitor” describes effects of ribociclib on various currents measured in GH3 cells. From their results, the authors conclude that “RIB-mediated perturbations in ionic currents presented herein is virtually upstream of its suppressive action on cytosolic CDK-4/6 activities and would partly participate in its modulatory effects on functional activities of pituitary tumor cells” (verbatim from the abstract).Unfortunately, these conclusions are not sufficiently supported by the data for the following reasons.
1) Currents shown in figure 6 cannot be M currents, as these currents activate slowly and do not show any inactivation during 1 s depolarizations. To verify that these currents were mediated by M channels (i.e. Kv7 channels), the authors would need to use appropriate blockers (e.g. XE991) and to apply ribociclib in the absence and presence of the chosen blocker.
2) The same holds true for the other types of currents reported in the manuscript. For each current type, ribociclib needs to be tested in the absence and presence of selective blockers to prove specificity of inhibitory actions.
3) In the discussion, one can read the following statement (p. 13, line 417): “Therefore, the concentration of RIB required for the inhibition of ionic currents present in GH3 and HL-1 cells appears to be mildly higher than that used to suppress the activity of CDK4/6.” Given the fact that IC50 values for CDK inhibition are in the range of 0.01 µM and IC50 values within the manuscript are 2.7 µM, this is hard to reconcile.
4) The authors claim that (p. 13, line416) “plasma concentration was currently reported to reach about 1036 ng/ml (i.e., about 2.38 µM)”. In this context, they forget to mention plasma protein binding of ribociclib in the range of 70 to 90 % (Bao et al, J Pharm Biomed Anal. 2019 Mar 20;166:197-204). Hence, free and active concentrations would be considerably lower. Moreover, most pharmacokinetic studies of ribociclib report plasma concentration below 1 µM.
5) The text is hard to understand because of problems regarding the English language; a few examples:
- 2, line 47: “CDK inhibitors (e.g., RIB) are thought to be known targets to become deregulated in 4different types of tumor cells including pituitary tumor cells”: how can a CDK inhibitor be a target and become deregulated?
- line 50: “CDK-4/6 inhibitors (e.g., RIB) have also been reported to protect the degenerative changes of motor neurons”: when degenerative changes are protected, what should be the therapeutic benefit?
- 8, line 273: “In the next stage of experiments, we wanted to examine assess if this drug has any perturbations on IK(DR) inherently in these cells”: what do the authors mean by examine assess?
Reviewer 2 Report
This manuscript evaluated pharmacological effects of a CDK-4/6 inhibitor, namely ribociclib, on several types of potassium channels in pituitary tumor cells and cardiac myocytes. The results suggested that this drug can affect the potassium currents in these cells. While most results look interesting, a few concerns including some key information remain to be addressed before this manuscript can be properly considered for publication.
1. In figure 2, 3, 8, I am not entirely convinced by the claim that these recorded currents were from Ik(erg) as the experimental protocol used could also stimulate other types of potassium currents such as Ito. The same issue also applies to the figure 4, 5 and 6 as well. Therefore validation of these currents with a reliable and specific pharmacological tool is necessary to make the conclusion solid.
2. In figure 2B and C, the presentation of drug effects on the potassium currents is very misleading. It should be changed to the one like Figure 8B. The currents before and after the addition of the drug are to be presented in same figure.
3. The colours for all these figures are very misleading, as each colour doesn’t represent a particular type of recordings consistently.
4. In figure 3, the example current trace for 3 micromolar RIB should also be shown in A.
5. In figure 4B, the current effect by RIB should include the one at the beginning and at the end, just like the one shown in C.
6. A 10 seconds depolarizing step was used in figure 5, though the experimental aim is almost same as figure 4. More explanation about why 10 seconds was chosen should be provided.
7. The biophysical kinetics provided in figure 5 for Ik(DR) should also be presented for other types of potassium currents shown in other figures.
8. One major concern remaining in current version of the manuscript is whether this compound exerted its effects through CDK-4/6 pathway in the cells. Therefore cell-detached recording configurations could be taken advantage of. Or the relationship of the time course versus the drug effect on the different types of potassium currents is to be provided to indicate how long it takes to get the pharmacological effect stable.